# Utilizing Computed Tomography to Analyze the Morphomic Change between Patients with Localized and Metastatic Renal Cell Carcinoma: Body Composition Varies According to Cancer Stage

**DOI:** 10.3390/jcm11154444

**Published:** 2022-07-30

**Authors:** Chin-Chieh Tan, Ting-Wen Sheng, Ying-Hsu Chang, Li-Jen Wang, Cheng-Keng Chuang, Chun-Te Wu, See-Tong Pang, I-Hung Shao

**Affiliations:** 1Department of Medicine, Chang Gung University, Taoyuan 333, Taiwan; jerrytan0307@gmail.com (C.-C.T.); changyinghsu@gmail.com (Y.-H.C.); ckchuang@gmail.com (C.-K.C.); wucgmh@gmail.com (C.-T.W.); pst64lab@gmail.com (S.-T.P.); 2Department of Medical Imaging and Intervention, New Taipei Municipal Tu Cheng Hospital, Chang Gung Memorial Hospital, Chang Gung University, New Taipei City 236, Taiwan; steven.sheng@gmail.com (T.-W.S.); lijenwang0918@gmail.com (L.-J.W.); 3Department of Urology, New Taipei Municipal Tu Cheng Hospital, Chang Gung Memorial Hospital, Chang Gung University, New Taipei City 236, Taiwan; 4Division of Urology, Department of Surgery, Chang Gung Memorial Hospital, Linkou Branch, Taoyuan 333, Taiwan; 5Division of Urology, Department of Surgery, Chang Gung Memorial Hospital, Keelong Branch, Keelung 204, Taiwan; 6Cancer Genome Research Center, Chang Gung Memorial Hospital, Linkou Branch, Taoyuan 333, Taiwan

**Keywords:** renal cell carcinoma, body composition, obesity, visceral fat, subcutaneous fat, skeletal muscle, computed tomography

## Abstract

Background: This study aimed to elucidate the change of body composition in different clinical stages of renal cell carcinoma (RCC) by analyzing computed tomography (CT) images. Methods: We enrolled patients diagnosed with RCC in a tertiary medical center who did not mention body weight loss or symptoms of cachexia. We grouped patients into those with localized RCC and those with metastatic RCC. Analyses of the volume of skeletal muscles tissue (SMT), subcutaneous adipose tissue (SAT), and visceral adipose tissue (VAT) calculated based on CT images were performed and included subgroup analyses by sex and age. The correlation between tumor size and body composition in localized RCC was also examined. Results: A total of 188 patients were enrolled in this study. There was significantly lower VAT (*p* = 0.015) in the metastatic group than in the localized group. SAT, body weight, and body mass index (BMI) were not significantly different between these two groups. In the subgroup analysis, a significant difference in SMT and VAT was noted in the male and younger subgroups but not in the female and older subgroups. Regarding primary tumor size in localized RCC, VAT was significantly higher in patients with larger tumors (*p* = 0.003). Conclusions: In localized RCC, VAT volume was significantly larger in those with large primary tumor size. However, the VAT was significantly lower in those with metastatic status comparing to those with localized disease. The clinical course of cancers closely correlates with body composition.

## 1. Introduction

Tumor growth is an energy-consuming process [1]. During the progression of tumors, appetite loss, starvation, anemia, poor immunity, unexpected weight loss, and other symptoms of cachexia will appear. With respect to body composition, loss of body weight and decreased skeletal muscle and adipose tissue mass appear in the cachexic state [2].

The kidney is an organ with an important role in metabolism, and tumors associated with the kidney may have a significant influence on metabolism. Cancer originating from the renal tubular epithelial cells is known as renal cell carcinoma (RCC) and accounts for up to 90% of cancers in the kidney [3]. RCC accounts for approximately 2% of the all cancer diagnoses worldwide [4]. There are three main subtypes of RCC: clear cell RCC (ccRCC), papillary RCC (pRCC), and chromophobe RCC (chRCC); ccRCC is the most common among them [5].

Body mass index (BMI), as a marker for obesity and body composition, is related to the risk of RCC. In previous observational studies, nearly half of all kidney tumors were linked to obesity (BMI > 30 kg/m^2^), and the risk of RCC is approximately 20–35% higher for every 5 kg/m^2^ increase in BMI [6]. In addition to BMI, the visceral adipocyte index has also been used to predict RCC and showed significant correlations with higher nuclear grade and tumor size [7]. In addition to the correlation with the occurrence of RCC, body composition itself can have an influence on the prognosis of RCC. Patients with advanced RCC who have higher amounts of visceral fat had a lower mortality rate after nephrectomy [8], while patients with metastatic RCC with high visceral fat area also had better prognosis after systemic therapy [9]. Skeletal muscle has been shown to have a great influence on the immune system; therefore, it has an influence on tumor progression [10]. The effect of skeletal muscle mass in patients with localized, locally advanced, and metastatic RCC have been investigated in some studies; however, the results have been inconclusive [11,12,13]. Cancer may have impact on body composition, while body composition may also potentially influence the outcome of cancers.

In this study, we measured the body composition variables of patients diagnosed with RCC based on computed tomography (CT) findings and aimed to elucidate the relationship between body composition factors and tumor status in RCC.

## 2. Materials and Methods

From 2007 to 2014, patients with newly diagnosed RCC in a tertiary medical center were retrospectively identified via electronic medical record and enrolled. Patients with a previous history of cancer or concomitant newly diagnosed cancer were excluded. Patients who had self-reported body weight loss, obvious poor appetite, or other cachexia-related symptoms that were documented on medical records were also excluded. All patients underwent CT of the abdomen for cancer staging. After discussion in our multi-disciplinary uro-oncology board, all patients received adequate treatment, including partial nephrectomy, radical nephrectomy, or systemic target therapy, according to their clinical condition.

This study was approved by the Chang Gung Medical Foundation Institutional Review Board. This study was conducted in accordance with the ethical principles mentioned in the Declaration of Helsinki (2013). The requirement for informed consent was waived due to the retrospective study design.

We collected patients’ general data by reviewing the medical records, including sex, age, height, weight, body mass index (BMI), hemogram, biochemistry laboratory data, and comorbidities. Tumor-related factors, such as TNM staging, histological subtypes, and tumor size, were also collected.

Body composition variables were measured based on the CT images for cancer staging. Abdominal computed tomography (CT) scans were performed with and without intravenous contrast before surgery as routine. The slice thickness and interval ranged from 3 to 10 mm with a median of 5 mm. Body composition analysis was performed using 3D Slicer (www.slicer.org, accessed on 1 January 2015 [14]) using a semiautomatic segmentation method. Abdominal CT images were segmented into three components: skeletal muscle tissue (SMT), subcutaneous adipose tissue (SAT), and visceral adipose tissue (VAT). The CT attenuation value of adipose tissue was defined as −190 to −30 Hounsfield units (HUs) [15]. Three-dimensional (3D) volumes at the level of the costophrenic angle to the iliac crest and two-dimensional (2D) cross-section areas at the level of the third lumbar spine (L3) showing both transversal processes were calculated for SMT, SAT, and VAT. Figure 1 shows an example of body composition analysis on abdominal CT images.

Independent *t*-test and descriptive statistics were performed using IBM SPSS v22.0 (IBM Corp., Armonk, NY, USA) to compare the difference between patients with localized RCC and metastatic RCC. Subgroup-analysis according to sex, age, and tumor size were also performed. The tumor size subgroup was divided by the median value of 3D tumor volume calculated on CT scan.

## 3. Results

Among all 188 patients, there were 141 patients with localized RCC and 47 patients with metastatic RCC; ccRCC was the most common histologic subtype (81.4%) among these patients. The mean age (±SD) of all patients was 57.1 ± 13.11 years, mean weight was 67.71 ± 12.68 kg, and mean BMI was 25.17 ± 4.01 kg/m^2^. The mean weight and BMI were not statistically different between the localized and metastatic groups (68.7 vs. 65.0 kg, *p* = 0.096; 25.4 vs. 24.4 kg/m^2^, *p* = 0.168, respectively). Other detailed clinical characteristics are listed in Table 1.

The mean SMT volume in the metastatic RCC group was 1735.28 cm^3^, which is significantly lower than that in the localized RCC group (mean value = 1948.49 cm^3^, difference = 231.21 cm^3^, *p* = 0.015). The VAT volume was significantly lower in the metastatic group than in the localized group (1523.2 cm^3^ vs. 1986.7 cm^3^, respectively, *p* = 0.015). The mean SAT volume was 1879.6 cm^3^ in the localized group and 1559.8 cm^3^ in the metastatic, which was not significantly different (*p* = 0.076).

The result of VAT cross sectional area analysis showed similar significant differences between the two groups. The mean area of VAT-L3 was 146.48 cm^2^ in the localized group and 118.52 cm^2^ in the metastatic group (*p* = 0.037). In multi-variate analysis, only VAT volume was significantly different between localized and metastatic subgroups (*p* = 0.020). Detailed results are listed in Table 2.

We then performed subgroup analysis by groups according to sex and age. In the sex subgroup analysis, SMT and VAT volume in metastatic RCC were significantly lower than in localized RCC in the male subgroup. The mean SMT volume was 2094.8 cm^3^ in the localized group and 1856.8 cm^3^ in the metastatic group in the male subgroup (*p* = 0.015), and the mean VAT volume was 2137.8 cm^3^ in the localized group and 1579.8 cm^3^ in the metastatic group (*p* = 0.012). However, the SMT and VAT volumes were not significantly lower in the metastatic group in the female subgroup analysis (*p* = 0.118 and 0.497, respectively).

In the age subgroup analysis, SMT volume was significantly lower in metastatic RCC than in localized RCC in the younger subgroup. A similar pattern was seen in the older subgroup; however, the difference was not significant. The detailed results are listed in Table 3, and the comparison of BMI and body composition factors in the whole group and different subgroups is illustrated in Figure 2.

The relationship between body composition factors and tumor burden was further examined. We divided patients in the localized RCC group by the median tumor volume (19.32 cm^3^) into small and large tumor groups.

For SMT and VAT, the mean volumes in the large tumor group were significantly higher than those in the small tumor group. However, only VAT remained significantly different in multi-variate analysis. The detailed results are listed in Table 4, while Figure 3 illustrates the difference in body composition factors between the large and small tumor groups.

## 4. Discussion

Loss of adipose tissue occurs earlier than loss of lean mass in patient with metastatic disease [16]. The change in body composition can occur before a noticeable change in BMI or net body weight. In our study, patients with late-stage RCC significantly had lower adipose tissue and lean mass than those with early-stage RCC, but there was no significant difference in body weight or BMI. We divided adipose tissue into VAT and SAT and found significantly lower VAT but not SAT in metastatic RCC, which might imply that VAT is consumed earlier than SAT during the clinical progression of RCC.

In contrast to SAT, VAT is more like to be the driver of metabolic disturbances in obesity [16]. Studies have shown that in cardiometabolic disease or acute cholecystitis VAT but not SAT is associated with disease outcomes [17,18]. A meta-analysis demonstrated that after engaging in exercise and going on a diet, taking weight-promoting drugs, or receiving bariatric surgery, the change in SAT is greater than that in VAT, but the proportional change in VAT is greater than that in SAT [19]. In our study, we observed a significant difference in VAT but not in SAT between patients with metastatic and localized RCC; this is compatible with the previous hypothesis that VAT, even if it is more life-threatening, is more vulnerable than SAT.

In univariate analysis, the SMT volume was also significantly decreased in patients with metastatic disease; however, the significance diminished in multi-variate analysis. Clinical and experimental research has identified mechanisms by which skeletal muscle is affected by tumor cells, such as protein synthesis or muscle cell autophagy [20]. The etiology of cancer-associated muscle wasting is multifactorial. Tumor metabolism requires fuel for energy and amino acids, and tumor-derived molecules will also elicit catabolic pathways at the tissue level in muscle. Moreover, endocrine, neural, and inflammatory derangements provide further catabolic drive [21]. The above metabolic change in patients with diffuse tumor burden agreed with our findings that SMT was lower in patients with metastatic RCC even if they were naïve to systemic therapy for advanced cancer.

In the subgroup analysis, the significant difference in SMT and VAT disappeared in the female and older subgroups. The baseline volume of skeletal muscle and visceral fat correlated with sex and age, and studies have shown that female and older subgroups tend to have small amounts of skeletal muscle and visceral fat [22,23,24]. In our study, we also noted that female and older patients had relatively low baseline SMT and VAT, which might explain the non-significant result in our subgroup analysis.

We further aimed to elucidate the correlation between body composition and localized renal tumors. We found the opposite result: that patients harboring tumors that were larger would have a significantly higher volume of SMT and VAT in univariate analysis when the RCC still remained a localized disease. However, the significance of SMT diminished again in multi-variate analysis.

Skeletal muscle has a great influence on the immune system, and those with sarcopenia show worse immune status [25]. It is possible that reactively increased skeletal muscle acts as a protective factor for the body to deal with tumor cells. Tumor progression depends on the interaction between tumor epithelial cells and surrounding stromal cells, which refers to peri-nephric visceral adipose tissue in RCCs. Visceral adipose tissue presents anatomical, cellular, and expression profiles different from those of subcutaneous adipose tissue [26]. Furthermore, the growth factors and cytokines secreted by adipose tissue may form a microenvironment suitable for tumor growth, which has a significant effect on the progression cancer [27,28]. In previous studies, increased visceral adipose tissue was noted in patients with both clear and non-clear cell renal cell carcinomas [29]. This hypothesis explains why increased SMT and VAT in patients with higher tumor burden are related to the body’s response to tumor progression, owing to either their immune reaction or the endocrine stimulation of the adipose tissue.

Our study has some limitations. First, the relatively small sample size might make it difficult to analyze the body composition differences in some subgroups, such as subgroups defined by tumor stage. Second, since we could not perform a cohort study to trace the change in body composition in patients diagnosed with RCC as tumor progressed, a case-control study design was conducted instead.

## 5. Conclusions

The body composition of patients in different stages of RCC differs. When the tumor is confined locally, the body tends to have increased VAT volume. However, in those with metastatic disease, VAT volume was significantly lower. This difference of body composition between patients with localized and metastatic disease was detectable earlier than the body weight and BMI. In addition to localized and metastatic status, the body composition also varied when considering different gender and age. The interaction between body composition and cancers required a further larger cohort and basic research studies to discover.

## Figures and Tables

**Figure 1 jcm-11-04444-f001:**
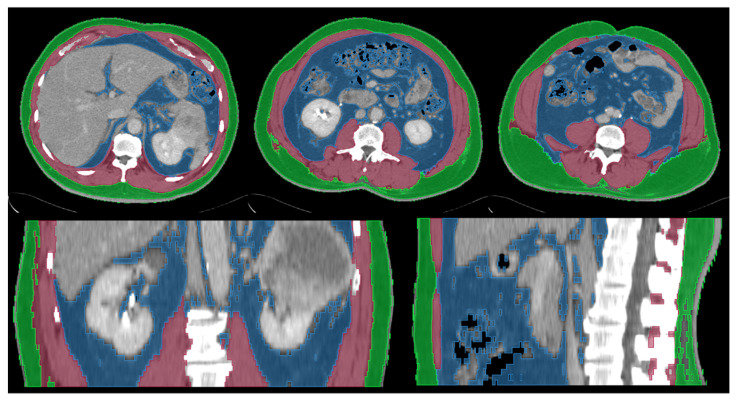
Example of abdominal computed tomography body composition analysis (**upper left**: axial plane at the level of costophrenic angle, **upper middle**: axial plane at the level of L3, **upper right**: axial plane at the level of iliac crest, **lower left**: coronal plane, **lower right**: sagittal plane). Skeletal muscles tissue (red area), subcutaneous adipose tissue (green area), and visceral adipose tissue (blue area) were segmented. Three-dimensional (3D) volumes from the level of costophrenic angle to iliac crest and two-dimensional (2D) cross-section areas at the level of L3 were calculated.

**Figure 2 jcm-11-04444-f002:**
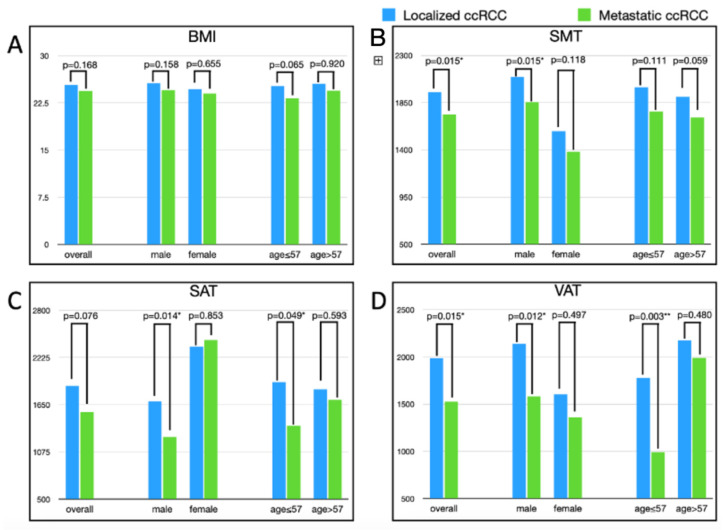
Comparison of body composition factors analysis between localized and metastatic RCC group in overall and subgroup (gender/age) analysis. (**A**) BMI; (**B**) SMT; (**C**) SAT; and (**D**) VAT. * *p*-value < 0.05; ** *p*-value < 0.01.

**Figure 3 jcm-11-04444-f003:**
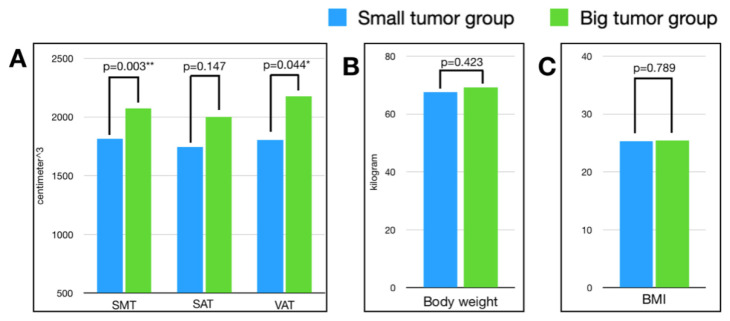
Comparison of body composition factors, body weight, and BMI in tumor size subgroup analysis in localized RCC. (**A**) SMT, SAT, and VAT; (**B**) body weight; and (**C**) BMI. * *p*-value < 0.05; ** *p*-value < 0.01.

**Table 1 jcm-11-04444-t001:** Patients’ General Characteristics.

Variables	Mean			SD	Range/ Percentage	
Age	57.1			13.11	11–82	Years old
Body weight	67.71			12.68	29–117	kg
Height	164.1			9.02	139.0–186.8	cm
BMI	25.17			4.01	15.01–39.55	
Tumor-Related Parameters						
Clinical T stage	1a		109	58.0%		
	1b		33	17.6%		
	2a		7	3.7%		
	2b		1	0.5%		
	3a		35	18.6%		
	3b		2	1.1%		
	3c		1	0.5%		
Clinical N stage	0		121	64.4%		
	1		29	15.4%		
	2		1	0.5%		
	x		35	18.6%		
Clinical M stage	0		141	75%		
	1		47	25%		
Tumor pathology	ccRCC	Stage I	107	72.3%		
		Stage II	2	1.4%		
		Stage III	11	7.4%		
		Stage IV	28	18.9%		
	pRCC	Stage I	13	76.5%		
		Stage II	0	0%		
		Stage III	0	0%		
		Stage IV	4	23.5%		
	chRCC	Stage I	12	66.7%		
		Stage II	0	0%		
		Stage III	0	0%		
		Stage IV	6	33.3%		
	mcRCC	Stage I	1	100%		
	Xp11.2 RCC	Stage I	2	100%		
	Other	Stage I	2	100%		
Body Composition Parameters						
SMT	1895.19			520.83	705.69–4148.16	cm^3^
SAT	1799.69			1070.70	5.61–7617.65	cm^3^
VAT	1870.86			1130.64	23.92–5725.61	cm^3^
SMT at L3	140.31			31.96	57.76–216.15	cm^2^
SAT at L3	138.02			70.65	38.1–254.4	cm^2^
VAT at L3	139.49			79.63	1.16–119.6	cm^2^

BMI, body mass index; ccRC, clear cell renal cell carcinoma; pRC, papillary renal cell carcinoma; chRC, chromophobe renal cell carcinoma; mcRC, multilocular cystic renal cell carcinoma. Xp11.2 RCC, Xp11.2 translocation renal cell carcinoma; SMT, skeletal muscle tissue; SAT, subcutaneous adipose tissue; VAT, visceral adipose tissue.

**Table 2 jcm-11-04444-t002:** Significant difference analysis between localized and metastatic RCC.

		Localized RCC (N = 141)		Metastatic RCC (N = 47)		Univariate Analysis		Multi-Variate Analysis
		Mean	SD	Mean	SD	Mean Difference	*p*-Value	*p*-Value
**Body weight**	kg	68.7	13.14	64.96	13.43	−3.74	0.096	0.799
**BMI**	kg/m^2^	25.36	4.11	24.39	4.27	−0.97	0.168	0.357
**SMT**	cm^3^	1948.49	511.89	1735.28	520	−213.21	0.015 *	0.981
**SAT**	cm^3^	1879.64	1038.50	1559.81	1140	−319.83	0.076	0.123
**VAT**	cm^3^	1986.74	1083.09	1523.2	1209.07	−463.54	0.015 *	0.020 *
**SMT-L3**	cm^2^	142.65	31.83	133.27	31.66	−9.38	0.082	0.471
**SAT-L3**	cm^2^	142.3	66.71	125.18	80.78	−17.12	0.151	0.573
**VAT-L3**	cm^2^	146.48	75.39	118.52	88.77	−27.96	0.037 *	0.736

SMT, skeletal muscle tissue; SAT, subcutaneous adipose tissue; VAT, visceral adipose tissue; SMT-L3, cross-section area of SMT at L3; SAT-L3, cross-section area of SAT at L3; VAT-L3, cross-section area of VAT at L3. * *p* value < 0.05.

**Table 3 jcm-11-04444-t003:** Subgroup Analysis Between Localized and Metastatic RCC.

		**Localized RCC**		**Metastatic RCC**			
		**(Male:Female = 101:40)**		**(Male:Female = 35:12)**			
		**Mean**	**SD**	**Mean**	**SD**	**Mean** **Difference**	***p*-Value**
**Body weight**	**Male**	72.29	12.18	66.65	13.16	−5.64	0.023 *
	**Female**	59.82	11.17	60.05	13.56	0.23	0.953
**BMI**	**Male**	25.63	3.9	24.53	4.07	−1.1	0.158
	**Female**	24.69	4.58	24	4.99	−0.69	0.655
**SMT**	**Male**	2094.8	487.26	1856.75	513	−238.09	0.015 *
	**Female**	1579	369.23	1381	365.33	−197.98	0.118
**SAT**	**Male**	1690.2	924.69	1259.32	751.6	−430.87	0.014 *
	**Female**	2358	1162.48	2436.23	1601.06	78.23	0.853
**VAT**	**Male**	2137.8	1070.63	1579.8	1254.89	−557.95	0.012 *
	**Female**	1605.5	1031.41	1358.13	1098.13	−247.34	0.497
**SMT-L3**	**Male**	156.09	25.62	143.51	28.71	−12.58	0.017 *
	**Female**	108.71	17.48	103.42	18.45	−5.29	0.39
**SAT-L3**	**Male**	133.75	62.58	106.88	54.8	−26.87	0.026 *
	**Female**	72.55	72.56	117.6	117.61	45.05	0.6
**VAT-L3**	**Male**	158.28	74.29	123.09	92.5	−35.19	0.025 *
	**Female**	70.58	70.58	79.01	79.01	8.43	0.656
				**Age subgroup**			
		**Localized RCC**		**Metastatic RCC**			
		**(Young:Old = 67:74)**		**(Young:Old = 22:25)**			
		**Mean**	**SD**	**Mean**	**SD**	**Mean** **Difference**	***p*-Value**
**Bodyweight**	**Young**	70.3	13.73	64.3	16.33	−6	0.094
	**Old**	67.25	12.5	65.55	10.57	−1.7	0.543
**BMI**	**Young**	25.2	4.12	23.23	4.76	−1.97	0.065
	**Old**	25.51	4.13	25.41	3.57	−0.1	0.92
**SMT**	**Young**	1996.5	565.92	1766.46	628.48	−230.08	0.111
	**Old**	1905	457.1	1707.85	413.5	−197.15	0.059
**SAT**	**Young**	1925.9	1080.34	1390.64	1142.58	−535.27	0.05
	**Old**	1837.8	1004.68	1708.68	1139.86	−129.07	0.593
**VAT**	**Young**	1778.5	1091.88	991.53	800.84	−786.99	0.003 **
	**Old**	2175.3	1046.91	1991.07	1324.23	−184.21	0.48
**SMT-L3**	**Young**	148.34	33.05	135.88	39.37	−12.46	0.147
	**Old**	137.5	29.99	130.98	23.54	−6.52	0.326
**SAT-L3**	**Young**	149.13	69.96	117.54	95.22	−31.59	0.098
	**Old**	136.1	63.46	131.9	66.88	−4.2	0.778
**VAT-L3**	**Young**	131.68	77.41	75.67	60.02	−56.01	0.003 **
	**Old**	159.88	71.41	156.22	93.74	−3.66	0.839

SMT, skeletal muscle tissue; SAT, subcutaneous adipose tissue; VAT, visceral adipose tissue; SMT-L3, cross-section area of SMT at L3; SAT-L3, cross-section area of SAT at L3; VAT-L3, cross-section area of VAT at L3. * *p*-value < 0.05; ** *p*-value < 0.01.

**Table 4 jcm-11-04444-t004:** Correlation Analysis Between Tumor Size and Body Composition in localized RCC.

		Small Tumor Group		Large Tumor Group		Univariate Analysis		Multi-Variate Analysis
		Mean	SD	Mean	SD	Mean Difference	*p*-Value	*p*-Value
**Body weight**	kilogram	67.59	12.35	69.18	13.05	−1.59	0.423	0.802
**BMI**	kg/cm^2^	25.28	4.12	25.44	4.02	−0.16	0.789	0.955
**SMT**	cm^3^	1815.22	478.37	2072.38	516.07	−257.16	0.003 **	0.319
**SAT**	cm^3^	1744.79	826.72	2001.81	1213.50	−257.02	0.147	0.866
**VAT**	cm^3^	1804.57	964.8	2176.96	1178.50	−372.39	0.044 *	0.003 **
**SMT-L3**	cm^2^	136.79	32.14	147.44	30.6	−10.65	0.047 *	0.109
**SAT-L3**	cm^2^	135.57	58.78	147.46	73.76	−11.89	0.295	0.914
**VAT-L3**	cm^2^	141.23	75.3	151.94	76.6	−10.71	0.407	0.002 **

SMT, skeletal muscle tissue; SAT, subcutaneous adipose tissue; VAT, visceral adipose tissue SMT-L3, cross-section area of SMT at L3; SAT-L3, cross-section area of SAT at L3; VAT-L3, cross-section area of VAT at L3. * *p*-value < 0.05; ** *p*-value < 0.01.

## Data Availability

All data generated or analyzed during this study are included in this published article.

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
