# Peer review of "Utilizing Computed Tomography to Analyze the Morphomic Change between Patients with Localized and Metastatic Renal Cell Carcinoma: Body Composition Varies According to Cancer Stage"

_jcm, 2022, doi:10.3390/jcm11154444_

Round 1

Reviewer 1 Report

In the manuscript " Utilizing Computed Tomography to Analyze the Morphomic Change Between Patients with Localized and Metastatic Renal Cell Carcinoma: The Role of Body Composition in Cancer Progression " , the authors examine the relationship between body composition as measured by skeletal muscles tissue (SMT), subcutaneous adipose tissue (SAT) and visceral adipose tissue (VAT) and the stage of tumor presentation. To measure these parameters, the authors used previously described and validated techniques with 3dslicer. 

Overall, it is an interesting study.  While other studies have looked at the impact of sketetal muscle tissue (SMT) and tumor stage, the relationship between overall body composition and RCC staging has not been fully investigated. 

1.  One point that I think that needs to clarified here.  When I read the word progression, I think of it as meaning change from one point to another.  Since these are all single point imaging, I think the authors should be using stage (local, locally advanced and metastatic disease).  It makes the point of manuscript unclear by using this word. 

INTRODUCTION

2.  page 2, sentence "The relationship between body composition and RCC progression should not be seen as cause-and-effect, and both of them have influences the other." I am unclear at to what the authors are trying to state here.  It seems like there is more that is meant to be said.   See point 1 regarding the word choice of progression. 

Materials and Methods

3.  How were the patients identified? From an EMR, trial, etc.  Please specify

4.  It is stated that "Patients who had self-reported body weight loss, obvious poor appetite, or other cachexia-related symptoms were also excluded."  Since this was a retrospective study, how was this determined?  I suspect through the electronic medical record

RESULTS

5.  Could the authors include in one of the tables a break down by tumor subtype and staging? 

6.  Table 2.  The title for the first column is "Localized ccRCC".  Is this correct that these were for the clear cell variant?  Or is it supposed to be all of the RCC? 

7.  Was there any significant difference between the stage, subtype of tumor and body composition? 

8.  Same issue regarding Table 3.  Was this analysis for clear cell RCC or all RCCs? 

9.  The authors state they divided the tumor size into "large" and "small" size to do a subanalysis of size.  But they don't provide any details about what was the sizes?  What was considered large?  Small?  How was it measured (transverse?  largest 3D dimension?)

DISCUSSION

10.  "With the metastasis of cancer, loss of adipose tissue occurs earlier than loss of lean mass [16]."  This is a confusing sentence, consider rewording such as "Loss of adipose tissue occurs earlier than loss of lean mass in patients with metastasis disease[16]"

CONCLUSION

11.  "The dynamic change in body composition in patients with different stages of RCC differs."  This sentence is confusing.  For example, consider using a different word than "change", it implies there are two different time points.

12.  "When the tumor is confined locally, the body responds to the increased tumor burden with in- creased SMT and VAT volume, possibly via an immune reaction or endocrine stimulation of the adipose tissue."  I am not sure how the authors can make this statement...  They don't have any data to support change over time.  

13.  Similarly, "However, as the tumor progress to metastatic disease, both SMT and VAT significantly decrease."  The authors are making the assumption that the body composition changes.  How do they know that baseline lower SMT and VAT is associated with increased risk of metastatic disease.  

14.  "As tumors progress, they affect body composition, and the body composition reacts to the tumor cell as well. Further larger cohort and basic research studies are required to investigate the interaction between body composition and cancers."  The data supports that patients with lower SMT and VAT are higher risk for metastatic disease.  They don't have temporal data to suggest there is clear link between tumor progression and body composition change, only body composition and stage of disease.  The authors will need temporal data linking tumor change to body composition. 

Author Response

1.One point that I think that needs to clarify here. When I read the word progression, I think of it as meaning change from one point to another. Since these are all single point imaging, I think the authors should be using stage (local, locally advanced and metastatic disease). It makes the point of manuscript unclear by using this word.

Thank you for the suggestion.

We have revised our manuscript and used staging instead of progression, in case of unclear meaning. We also revised the subtitle as “The Body Composition Differ in Different Stage of Cancer”.  

2.page 2, sentence "The relationship between body composition and RCC progression should not be seen as cause-and-effect, and both of them have influences the other." I am unclear at to what the authors are trying to state here. It seems like there is more that is meant to be said. See point 1 regarding the word choice of progression.

We had revised the statement as below: “Cancer may have impact on body composition, while body composition may also influence the outcome of cancers.”

3.How were the patients identified? From an EMR, trial, etc. Please specify

Patients was retrospectively identified from EMR and enrolled. We had revised the statement to specify this in the Materials and Methods.

4.It is stated that "Patients who had self-reported body weight loss, obvious poor appetite, or other cachexia-related symptoms were also excluded." Since this was a retrospective study, how was this determined? I suspect through the electronic medical record

We retrospectively reviewed the EMR and excluded those who had documented self-reported body weight loss, obvious poor appetite, or other cachexia-related symptoms on EMR. We had revised this statement in the manuscript to make it clear.

5.Could the authors include in one of the tables a break down by tumor subtype and staging?

Thank you for the advice. We had added this in table 2.

6.Table 2. The title for the first column is "Localized ccRCC". Is this correct that these were for the clear cell variant? Or is it supposed to be all of the RCC?

It is supposed to be all of the RCC. We had made the correction.

7.Was there any significant difference between the stage, subtype of tumor and body composition?

Since ccRCC accounted for most of the histology subtype, the analysis was not performed due to too small sample size of other subtypes. However, we still listed the percentage of different stage of each subtype in table 2.

8.Same issue regarding Table 3. Was this analysis for clear cell RCC or all RCCs?

It is supposed to be all of the RCC. We had made the correction.

9.The authors state they divided the tumor size into "large" and "small" size to do a subanalysis of size. But they don't provide any details about what was the sizes? What was considered large? Small? How was it measured (transverse? largest 3D dimension?)

The size was divided by the median value of 3D tumor volume calculated on CT scan. We had added the description to specify this in Materials and Methods.

10."With the metastasis of cancer, loss of adipose tissue occurs earlier than loss of lean mass [16]." This is a confusing sentence, consider rewording such as "Loss of adipose tissue occurs earlier than loss of lean mass in patients with metastasis disease [16]"

Thank you for the advice. We had made correction in Conclusion section.

11."The dynamic change in body composition in patients with different stages of RCC differs." This sentence is confusing. For example, consider using a different word than "change", it implies there are two different time points.

Thank you for the advice.  We had made correction in Conclusion section.

12."When the tumor is confined locally, the body responds to the increased tumor burden with in- creased SMT and VAT volume, possibly via an immune reaction or endocrine stimulation of the adipose tissue." I am not sure how the authors can make this statement... They don't have any data to support change over time.

Thank you for the advice. To make the expression of sentence more properly in Conclusion section, we revised the conclusion with more objective statement as below:

The body composition of patients in different stages of RCC differs. When the tumor confined locally the body tends to have increased VAT volume. However, in those with metastatic disease, VAT volume was significantly lower. This difference of body composition between patients with localized and metastatic disease was detectable earlier than the body weight and BMI. In addition to localized and metastatic status, the body composition also varied when considering different gender and age. The interaction between body composition and cancers required further larger cohort and basic research studies to discover.

13.Similarly, "However, as the tumor progress to metastatic disease, both SMT and VAT significantly decrease." The authors are making the assumption that the body composition changes. How do they know that baseline lower SMT and VAT is associated with increased risk of metastatic disease.

Thank you for the advice. To make the expression of sentence more properly in Conclusion section, we revised the conclusion with more objective statement as below:

The body composition of patients in different stages of RCC differs. When the tumor confined locally the body tends to have increased VAT volume. However, in those with metastatic disease, VAT volume was significantly lower. This difference of body composition between patients with localized and metastatic disease was detectable earlier than the body weight and BMI. In addition to localized and metastatic status, the body composition also varied when considering different gender and age. The interaction between body composition and cancers required further larger cohort and basic research studies to discover.

14."As tumors progress, they affect body composition, and the body composition reacts to the tumor cell as well. Further larger cohort and basic research studies are required to investigate the interaction between body composition and cancers." The data supports that patients with lower SMT and VAT are higher risk for metastatic disease. They don't have temporal data to suggest there is clear link between tumor progression and body composition change, only body composition and stage of disease. The authors will need temporal data linking tumor change to body composition.

Thank you for the advice. To make the expression of sentence more properly in Conclusion section, we revised the conclusion with more objective statement as below:

The body composition of patients in different stages of RCC differs. When the tumor confined locally the body tends to have increased VAT volume. However, in those with metastatic disease, VAT volume was significantly lower. This difference of body composition between patients with localized and metastatic disease was detectable earlier than the body weight and BMI. In addition to localized and metastatic status, the body composition also varied when considering different gender and age. The interaction between body composition and cancers required further larger cohort and basic research studies to discover.

Reviewer 2 Report

The aim of the study was to evaluate relationship between body composition and the progression of renal cell carcinoma, analyzing CT images. Study is interesting even if there are some critical aspects that Authors need to review in order to improve overall quality of manuscript. 

- Title should be reviewed. Body composition has not any role in cancer progression, on the contrary cancer progression should play a significant role in body composition. 

- In table 2, Authors properly considered BMI and body weight as a continuos variables. Since a trend towards significant could be detected in body weight (p=0.096) and BMI (p=0.168), has an analsysis been tried considering them categorical variables (for example: obese versus not obese; body weight >70 kg etc.). 

- Subgroup analysis has properly performed. However univariable and multivariable analysis should per performed, to better describe correlation of variables and several confounders. 

Author Response

The aim of the study was to evaluate relationship between body composition and the progression of renal cell carcinoma, analyzing CT images. Study is interesting even if there are some critical aspects that Authors need to review in order to improve overall quality of manuscript.

Title should be reviewed. Body composition has not any role in cancer progression, on the contrary cancer progression should play a significant role in body composition.

:Thank you for the suggestion. We had revised the title as below:

Utilizing Computed Tomography to Analyze the Morphomic Change Between Patients with Localized and Metastatic Renal Cell Carcinoma: The Body Composition Differ in Different Stage of Cancer

In table 2, Authors properly considered BMI and body weight as a continuos variables. Since a trend towards significant could be detected in body weight (p=0.096) and BMI (p=0.168), has an analsysis been tried considering them categorical variables (for example: obese versus not obese; body weight >70 kg etc.).

Thank you for the advice.

In this article, we did not categorized patients by body weight or BMI for some reasons. First, we focused more on the body composition instead of body weight and BMI, because we believed that the body composition can better represent body status than body weight or BMI alone. Second, neither the body weight/BMI revealed statistical significance in univariate and multi-variate analysis.

Subgroup analysis has properly performed. However univariable and multivariable analysis should performed, to better describe correlation of variables and several confounders.

Thank you for the suggestion. We had performed multi-variate analysis in localized/metastatic subgroup and tumor size subgroup and revised the table 2 and table 4.

Reviewer 3 Report

A simple topic. not relevant to the currently existing literature.Nothing to add to the readers interest and does not seem to be sounded scientfically.

Patients were enrolled between 2007-2014, average 10 years ago.

Author Response

A simple topic. not relevant to the currently existing literature.  Nothing to add to the readers interest and does not seem to be sounded scientfically.

As far as we know, we did not find any other same survey. 

Patients were enrolled between 2007-2014, average 10 years ago.

The data was built based on EMR.

Round 2

Reviewer 2 Report

No further comments. Authors properly addressed reviewer's comments.

Author Response

Thank you for your comments.